# Evidence-Gated Scientific QA with Explicit Abstention and Page-Level Provenance

**Bruno Leonardo Santos Menezes**[*]
National Laboratory for Scientific Computing (LNCC)
Petrópolis, RJ, Brazil
`brunolsm@lncc.br`

**Fábio Porto**
National Laboratory for Scientific Computing (LNCC)
Petrópolis, RJ, Brazil
`fporto@lncc.br`

## Abstract

Large Language Models have demonstrated strong performance in scientific question-answering tasks, particularly when combined with retrieval-based mechanisms. However, in high-risk scientific domains, reliability depends not only on access to external knowledge, but on the system's ability to determine when answering a question is epistemically justified. Existing Retrieval-Augmented Generation pipelines primarily address knowledge access, but lack explicit decision policies governing when generation should be authorized or withheld under insufficient evidence. In this work, we introduce Pororoca, an Evidence-Gated Scientific QA system that treats question answering as a system-level decision problem. Pororoca conditions generation on the explicit sufficiency of verifiable scientific evidence and enforces abstention otherwise, producing only answers accompanied by auditable provenance at the document and page level. The system operates on a scientific corpus automatically structured by a large-scale Document AI pipeline and implements a deterministic, threshold-based decision policy separating conditional generation from explicit abstention. We describe the system architecture, decision logic, and an epistemically auditable evaluation protocol designed to assess evidence-based factuality, citation quality, and selective risk under realistic retrieval noise. By framing scientific QA reliability as a property of explicit decision policy rather than model behavior alone, this work contributes a principled system-level approach to verifiable and reliable scientific question answering.

## 1 Introduction

Large Language Models (LLMs) have achieved strong performance in scientific question answering; however, in high-risk scientific domains, usefulness depends not only on fluency but on epistemic reliability the ability to generate answers supported by verifiable evidence. Retrieval-Augmented Generation (RAG) advanced this goal by grounding generation in external document collections, mitigating parametric knowledge limitations and becoming the foundation of modern scientific QA systems (Lewis et al., 2020).

However, RAG alone does not ensure scientific reliability. Even with retrieval, models may generate fluent responses under weak or absent evidence, underscoring the gap between linguistic fluency and epistemic validity. Prior work, notably SQuAD 2.0, formalized the importance of unanswerable questions, showing that knowing when not to answer is as critical as answering correctly (Rajpurkar et al., 2018). Subsequent work has shown that retrieval does not impose a decision policy on generation. Models may combine partially relevant snippets, extrapolate beyond available evidence, or incorporate retrieval noise. Improvements in multi-hop QA only emerge when generation is explicitly conditioned on the structure of the retrieved evidence, not merely on its presence (Fang et al., 2024). Thus, while RAG expands information availability, it does not enforce the criterion of epistemological sufficiency required for reliable generation.

Recent studies reinforce this conclusion by showing that scientific QA systems must distinguish between corroborating and refuting evidence. Integrated verification frameworks demonstrate that

---

[*]Correspondence: `brunolsm@lncc.br`

conventional RAG pipelines fail to evaluate evidence consistency and sufficiency, indicating that retrieval alone is insufficient in high-stakes scientific settings (Wang et al., 2025). The literature further emphasizes the need for explicit and auditable provenance. Benchmarks such as KILT argue that answers should be evaluated by their ability to point to traceable source evidence (Petroni et al., 2021). Attributable generation studies show that superficial citations do not guarantee factual support, reinforcing the need to tightly align generation with evidence (Menick et al., 2022). Selective generation formalizes the decision to respond versus abstain as a core system problem, showing that reducing factual errors requires explicit abstention mechanisms and control of the risk–coverage trade-off (Thakur et al., 2024). Factuality metrics such as FActScore advance this direction by evaluating responses solely based on cited evidence (Min et al., 2023). Empirical benchmarks confirm that, even with retrieval, the absence of explicit decision policies leads to persistent hallucinations (Liu et al., 2023). These findings converge on a central question: how to align scientific QA outputs with factual evidence without hallucination. Existing approaches primarily address this at the level of model training, calibration, or post-hoc verification. In contrast, we argue that hallucination in scientific QA is fundamentally a system-level decision problem. Reliable alignment therefore requires an explicit mechanism that determines when generation is epistemically justified and when it is not. Pororoca operationalizes this principle through an evidence-gated decision policy that conditions generation on the sufficiency of verifiable scientific evidence and enforces explicit abstention otherwise. In this formulation, factual alignment is not an emergent model property but a constraint enforced by system policy and validated under realistic retrieval and scoring noise. These observations motivate the following thesis:

> Scientific reliability in QA requires explicit decision policies that determine when to respond and when to refrain, conditioning generation on the existence of verifiable evidence.

Based on this thesis, we present Pororoca, an Evidence-Gated Scientific QA system that integrates explicit abstention, auditable evidence thresholds, and document- and page-level provenance. Unlike approaches that treat RAG as a sufficient solution, Pororoca frames operational reliability as a property of system policy rather than model capability. The main contributions of this work are:

- C1: An evidence-gated generation policy that conditions generation on explicit evidence sufficiency.
- C2: An explicit abstention mechanism for scenarios lacking adequate scientific support.
- C3: Verifiable provenance at the document and page level, enabling direct human auditing.
- C4: A selective evaluation protocol combining evidence-based factuality, citation quality, and risk–coverage analysis.

This framework positions Pororoca as a methodological contribution to reliable scientific QA in high-risk domains, aligned with epistemological norms of scientific practice and recent work on verifiable generation.

## 2 Pororoca: Evidence-Gated Decision Policy for Scientific QA

Pororoca is formulated as an explicit decision policy for scientific question answering, designed to align large language models with the epistemological norms of scientific practice. Unlike RAG systems treated as linear retrieval–generation pipelines, Pororoca frames generation as a decision-making process conditioned on the sufficiency of verifiable scientific evidence. Its core principle is that, in the absence of adequate evidence, the system must abstain from responding. This is enforced through an evidence-gated policy that deterministically chooses between conditional generation and explicit abstention, authorizing only responses accompanied by auditable document- and page-level provenance. The system operates on a scientific corpus structured by a large-scale Document AI pipeline that collects, extracts, normalizes, and indexes PDF articles, which constitutes an operational prerequisite. Figure 1 illustrates the resulting evidence-gated decision flow, abstracting implementation details to highlight the epistemological logic governing generation authorization and abstention.

Given a scientific question $q$, Pororoca constructs an explicit set of evidence $\mathcal{E}(q) = \{e_1, \ldots, e_k\}$ through vector retrieval followed by cross-ranking. Each piece of evidence corresponds to a local-

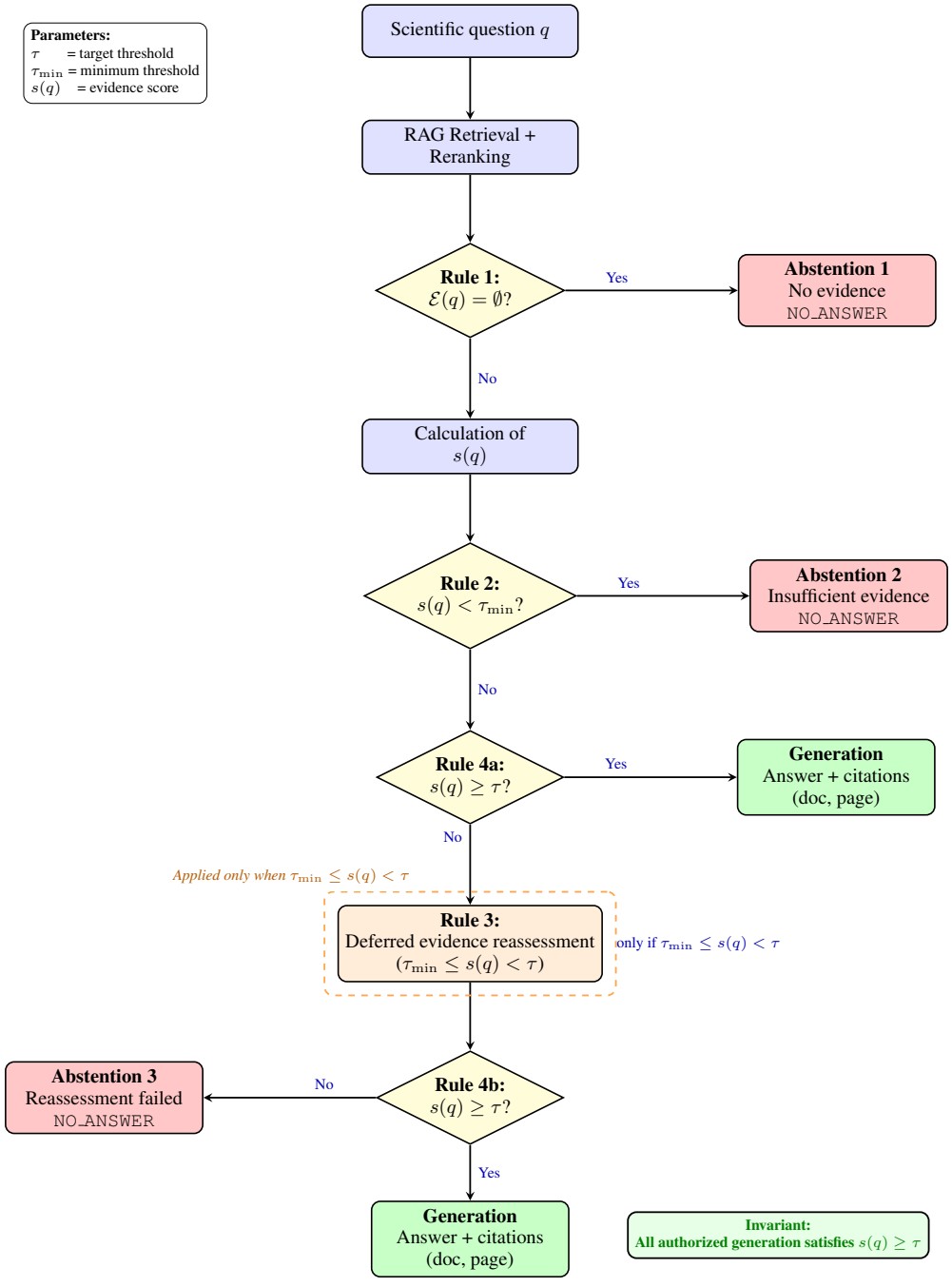

Figure 1: Flowchart of the Pororoca system's evidence-gated decision policy. The system performs explicit abstention when there is no evidence or when it is below the epistemological threshold, applying deferred evidence reassessment only when $\tau_{\min} \leq s(q) < \tau$. Every generation is authorized exclusively when $s(q) \geq \tau$, with the sufficiency condition checked both immediately after initial scoring (Rule 4a) and after potential refinement in the reassessment stage (Rule 4b), ensuring, by construction, the elimination of responses under insufficient evidence.

ized excerpt, associated with a specific document and page. The retrieved evidence is not treated as auxiliary context, but as the only explicitly provided and auditable input for generating responses. This explicit contract between evidence and generation ensures traceability, verifiability, and alignment with scientific practices.

Let $\text{score}(e_i, q) \in [0, 1]$ denote the relevance score assigned to each evidence excerpt $e_i$ by the cross-encoder reranker. The aggregate evidence score $s(q)$ is defined as the maximum score over the top-$K$ retrieved passages after reranking:

$$s(q) = \max_{i=1,\ldots,k} \text{score}(e_i, q),$$

where $k = |\mathcal{E}(q)|$ is the number of retrieved passages. This max-aggregation is intentionally conservative: it reflects the strength of the single best piece of evidence available, rather than an average over potentially noisy or partially relevant passages. A passage qualifies for inclusion in $\mathcal{E}(q)$ only if its score exceeds a minimum admissibility threshold $\tau_{\min}$, so the final evidence set consists exclusively of passages with $\text{score}(e_i, q) \geq \tau_{\min}$.

The two thresholds $\tau$ and $\tau_{\min}$ are selected on a held-out validation split of the evaluation dataset, distinct from the test set used for all reported results. Candidate threshold pairs are swept over the grid $\tau_{\min} \in \{0.05, 0.10, 0.15, 0.20\}$ and $\tau \in \{0.20, 0.25, 0.30, 0.35, 0.40\}$ (with $\tau > \tau_{\min}$ enforced), and the pair that maximises abstention accuracy on the validation split is selected. For all experiments reported in this paper, the selected values are $\tau = 0.30$ and $\tau_{\min} = 0.10$.

When $\tau_{\min} \leq s(q) < \tau$ (Rule 3 in Figure 1), the system enters a reassessment stage. In this stage, the evidence set $\mathcal{E}(q)$ is re-examined by expanding the retrieval to a broader candidate pool (increasing $k_1$ from 50 to 100 top FAISS candidates) and re-scoring the expanded set with the cross-encoder. The aggregate score $s(q)$ is then recomputed on this refined set. If the revised score satisfies $s(q) \geq \tau$, generation is authorized (Rule 4b); otherwise the system abstains. Importantly, the reassessment does not alter the decision thresholds: $\tau$ and $\tau_{\min}$ remain fixed throughout. Its sole purpose is to obtain a more precise evidence estimate in borderline cases before committing to a final deterministic decision.

The evidence-gated policy is parameterized by two auditable thresholds: $\tau$, target threshold of sufficiency, and $\tau_{\min}$, minimum threshold of epistemological admissibility.

The system's decision follows a deterministic, multi-stage decision rule:

1. If $\mathcal{E}(q) = \emptyset$, explicit abstention.

2. If $s(q) < \tau_{\min}$, explicit abstention.

3. If $\tau_{\min} \leq s(q) < \tau$, deferred evidence reassessment, followed by a final sufficiency check.

4. If $s(q) \geq \tau$, conditional generation (checked at each authorization point in the flow).

The condition $s(q) \geq \tau$ is evaluated whenever the system reaches a generation authorization point in the decision flow. The reassessment stage does not modify the sufficiency threshold $\tau$, but allows additional aggregation or verification of evidence before a final deterministic decision is made, potentially refining the evidence representation used to compute $s(q)$ without altering the decision policy itself. The initial score $s(q)$ is intentionally conservative; the reassessment stage allows a more precise estimate in borderline cases without relaxing the epistemic sufficiency criterion. Abstention is treated as a correct decision in scenarios of insufficient evidence, rather than as a model failure. Determinism here refers to the decision policy applied to a fixed retrieval and scoring outcome; given identical evidence and scores, the resulting decision is fully determined by the policy thresholds. When authorized, generation occurs exclusively based on $\mathcal{E}(q)$. The response is accompanied by structured citations at the document and page level, allowing for immediate human verification. When policy dictates abstention, the system returns NO_ANSWER, avoiding the production of fluently formulated, but epistemologically unjustified responses.

## 2.1 ISOLATION OF PARAMETRIC KNOWLEDGE

To ensure that generated responses are conditioned exclusively on retrieved scientific evidence, Pororoca implements the following mechanisms:

1. Explicit prompting strategy: The generation prompt explicitly instructs the LLM to base responses solely on the provided evidence excerpts, with strict instructions against the use of external or parametric knowledge.

2. Evidence-only context window: During generation, the model receives only the retrieved evidence set $\mathcal{E}(q)$ as context, with no access to the broader document collection or external knowledge sources.

3. Citation requirement: All generated statements must be accompanied by explicit citations pointing to specific evidence excerpts, creating a traceable chain from answer to source.

4. Post-generation verification: A verification step checks that all factual claims in the generated answer are supported by at least one cited evidence excerpt, filtering out any unsupported statements.

While we acknowledge that complete elimination of parametric influence is not formally verifiable, these mechanisms establish strong operational constraints that align generation with the evidence-gated policy.

## 3 EXPERIMENTS AND EVALUATION

The objective of the experimental evaluation is to assess the operational reliability of the Pororoca system in realistic scientific scenarios. We seek to measure the system's ability to (i) respond only when there is sufficient scientific evidence, (ii) explicitly refrain from responding in the absence of adequate evidence, and (iii) provide responses accompanied by verifiable provenance at the document and page level. The experimental protocol was designed to be epistemically auditable end to end: all evidence availability labels, abstention decisions, and evaluation metrics are fully traceable to explicit corpus content and logged system actions. The same Pororoca system is used both for (i) the structured extraction of the scientific corpus and for (ii) the evaluation of decision policies during inference. The differences between the systems evaluated lie exclusively in the generation or abstention policies, which allows for a controlled and interpretable comparison of epistemic decision policies under identical retrieval and generation components. The construction of the evaluation set follows a consolidated approach in the literature, in which deliberately unanswerable questions are included to assess the system's ability to recognize the absence of evidence and avoid hallucination (Rajpurkar et al., 2018). Unlike generic benchmarks, our set is derived directly from the target scientific corpus, preserving semantic alignment and traceability. First, scientific articles in PDF format are processed by the Pororoca system itself, which incorporates olmOCR (Poznanski et al., 2025), a 7B vision language model fine-tuned for extracting clean, linearized plain text from PDFs while preserving structured content such as sections, tables, lists, and equations. This extraction process produces auditable `.jsonl` files with structured text and metadata at the page level. Next, these JSONLs are used as input for an LLM (provided by OpenAI), employed exclusively for the linguistic formulation of questions in the field of meteorology. The generation process operates as follows: for each document in the corpus, the system extracts structured content at the page level and provides it as context to the LLM, which then generates candidate questions based solely on the explicitly provided textual excerpts. The LLM does not perform retrieval, does not assess evidence sufficiency, and does not assign labels—it serves only as a linguistic generator conditioned on corpus content. This procedure follows approaches for automatic question generation from large document collections (Lewis et al., 2021). Each question is labeled `has_evidence` or `no_evidence` based on explicit corpus support, yielding a balanced set of 1020 questions (510 per class). This design enables controlled evaluation of both grounded answer quality and abstention decisions, consistent with knowledge-intensive QA benchmarks requiring explicit provenance (Petroni et al., 2021). Annotation guidelines, evidence availability criteria, and human validation protocols are detailed in Appendix A.

Before final incorporation into the evaluation corpus, all automatically generated questions and their labels (`has_evidence` / `no_evidence`) underwent human validation by technically trained annotators, who explicitly verified evidence availability at the document and page level. This follows established practices in robust QA and verifiable generation (Rajpurkar et al., 2018), where human annotation ensures semantic reliability and avoids spurious labels. The requirement for traceable evidence and explicit ground truth aligns with knowledge-intensive benchmarks requiring verifiable provenance (Petroni et al., 2021), and factual evaluation based exclusively on cited evidence follows the formalism of recent metrics like FActScore (Min et al., 2023). Only questions whose labeling showed full human agreement were incorporated into the final set, resulting in a balanced dataset with 1020 instances. The protocol followed by human annotators, as well as the operational

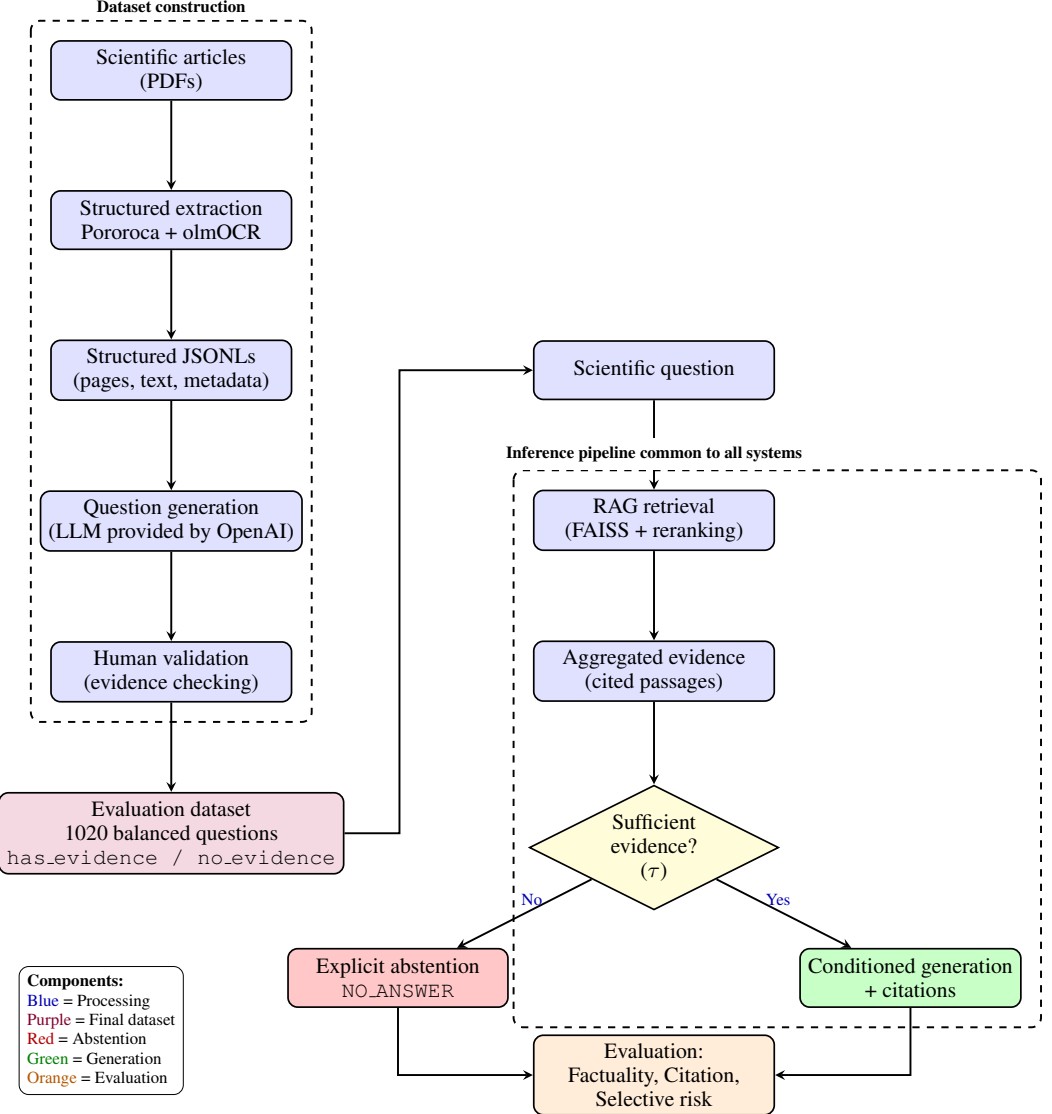

Figure 2: Complete experimental pipeline of the Pororoca system. Dataset construction (left) uses the system itself for structured extraction via olmOCR (Poznanski et al., 2025), followed by automatic question generation and human validation, resulting in 1020 balanced questions. The inference pipeline (right) is common to all evaluated systems, with differences only in the decision policy at the evidence gate. Evaluation measures evidence-based factuality, citation quality, and selective risk. This design isolates epistemic decision policies as the sole experimental variable.

definition of sufficient evidence at the document and page level, are formalized in Appendix A. This dataset feeds the inference pipeline, in which all evaluated systems share exactly the same retrieval, reranking, and generation mechanisms. The only point of variation between baselines is the evidence gate, parameterized by a threshold $\tau$, which decides between explicit abstention and conditional generation. This explicit separation allows us to isolate the effect of the system's decision policies and measure, in a controlled manner, (i) the coverage of the generated responses, (ii) the factual risk associated with the responses provided, and (iii) the accuracy of the decision to abstain. Next, we formalize these quantities through explicitly defined metrics, aligned with the formalism of selective prediction and evidence-based generation.

## 3.1 METRICS

Pororoca's evaluation is guided by scientific reliability and follows metrics explicitly formulated to verify not only factual correctness, but also attribution, provenance, and selective behavior. This approach is aligned with recent work arguing that question-answering systems based on LLMs should generate responses only when supported by verifiable evidence, avoiding hallucination through explicit abstention (Menick et al., 2022; Gao et al., 2023). All metrics are computed exclusively from the retrieved evidence and explicit decisions of the system. The generation process is designed to minimize reliance on parametric knowledge through explicit prompting constraints and citation requirements, as described in Section 2.2.

### 3.1.1 EVIDENCE-BASED FACTUALITY

For each answered instance $(a_i, \mathcal{E}_i)$, the same three annotators independently assign a support category. Full support ($\texttt{support} = 1$) requires that every factual claim in $a_i$ is directly verifiable from at least one excerpt in $\mathcal{E}_i$, without requiring inference or external knowledge. Partial support ($\texttt{support} = \alpha = 0.5$) is assigned when some claims are directly supported but others require minor inference. No support ($\texttt{support} = 0$) is assigned when no claim can be traced to $\mathcal{E}_i$. The inter-annotator agreement for support categorisation is $\kappa = 0.79$. Disagreements are resolved by majority vote.

The average factuality score of the system is given by:

$$\text{Factuality} = \frac{1}{|S|} \sum_{i \in S} \text{support}(a_i, \mathcal{E}_i),$$

where $S$ denotes the subset of instances for which the system authorizes answer generation. Instances resulting in explicit abstention ($\texttt{NO\_ANSWER}$) are excluded from factuality computation and evaluated separately through abstention accuracy and selective risk metrics, following the spirit of metrics based on explicit attribution, such as FActScore (Min et al., 2023).

### 3.1.2 QUALITY OF CITATION AND ATTRIBUTION

To evaluate the quality of the source, each pair $(a_i, \mathcal{E}_i)$ is classified into three categories:

$$\text{CitationQuality}(a_i) \in \{\text{GOOD}, \text{WEAK}, \text{BAD}\}.$$

A citation is considered GOOD if the cited evidence contains sufficient information to verify the answer without external inference. It is considered WEAK if the relationship is indirect or incomplete, and BAD if there is no verifiable support. This classification is performed by the same three annotators who labelled evidence availability. Annotators apply the categories independently per instance; inter-annotator agreement for citation quality is $\kappa = 0.76$, with disagreements resolved by majority vote. We report the empirical distribution of these categories, in line with recent practices in verifiable generation evaluation and explicit attribution (Liu et al., 2023; Gao et al., 2023).

### 3.1.3 ACCURACY OF ABSTENTIONS

The decision to abstain is treated as a binary classification problem. Let $y_i \in \{0, 1\}$ be the evidence label of question $i$ ($0 = \texttt{has\_evidence}$, $1 = \texttt{no\_evidence}$) and $\hat{y}_i$ the system's decision ($1 = $ abstain, $0 = $ answer). Here the *positive class* is $\texttt{no\_evidence}$: a true positive is a question correctly identified as lacking sufficient evidence (correct abstention), and a false negative is a question lacking evidence that the system answers (incorrect generation). The accuracy of abstention is defined as:

$$\text{AbstentionAccuracy} = \frac{1}{N} \sum_{i=1}^{N} \mathbb{I}[\hat{y}_i = y_i],$$

Table 1: Evidence-based factuality computed over answered instances only.

| System | Factuality | #Answered |
|--------|-----------|-----------|
| baseline_always | 0.798 | 233 |
| rag_no_abstention | 0.798 | 233 |
| pororoca | 0.789 | 246 |

where $\mathbb{I}[\cdot]$ is the indicator function. We also report precision and recall of the abstention decision, alongside a full confusion matrix (Table 4).

### 3.1.4 SELECTIVE RISK CURVES

To analyze the trade-off between coverage and reliability, we follow the classical formalism of selective prediction. Let $\tau$ be an evidence threshold and $S_\tau$ the set of questions answered under that threshold. We define:

$$\text{Coverage}(\tau) = \frac{|S_\tau|}{N},$$

$$\text{Risk}(\tau) = \frac{1}{|S_\tau|} \sum_{i \in S_\tau} \mathbb{I}[\text{support}(a_i, \mathcal{E}_i) = 0].$$

The risk-coverage curve is obtained by varying $\tau$, allowing us to explicitly visualize the selective behavior of the system, as discussed in Thakur et al. (2024).

## 4 RESULTS

We compare three system configurations that share an identical document corpus, retrieval pipeline, reranking mechanism, and language model (**Qwen-7B-Chat**, greedy decoding, max_new_tokens = 768). Retrieval uses FAISS ($k_1$ = 50) followed by cross-encoder/ms-marco-MiniLM-L-6-v2 selecting the top-12 passages. The systems differ *exclusively* in their decision policies:

- baseline_always: always generates an answer; falls back to parametric LLM knowledge when retrieval returns no passages. No abstention capability.

- rag_no_abstention: applies full retrieval and reranking but no score threshold. When passages are available, generation is always authorized regardless of scores; when retrieval fails, produces a context-free response without parametric fallback. Both baselines apply no threshold, which is why they yield identical results on the 233 questions where retrieval succeeds— they diverge only on zero-retrieval cases (full operational details in Appendix B).

- pororoca: applies the evidence-gated policy with $\tau = 0.30$ and $\tau_{\min} = 0.10$.

This controlled setup isolates epistemic decision-making as the sole experimental variable. All results are computed directly from execution logs, ensuring full auditability (Appendix B). We first evaluate evidence-based factuality, computed only over instances where a system authorizes answer generation. As shown in Table 1, all systems exhibit comparable factuality scores, with Pororoca achieving a factuality of 0.789, closely matching both baselines. This result is expected by construction. When Pororoca authorizes generation, it does so exclusively under the condition that sufficient evidence exists ($s(q) \geq \tau$), ensuring that the factual quality of authorized answers is not degraded relative to systems that always respond. The similarity of factuality scores confirms that the proposed decision policy does not trade correctness for conservatism.

We next analyze citation quality, classifying each generated answer as GOOD, WEAK, or BAD based on the sufficiency of its cited evidence. Table 2 reports the empirical distribution. Across all systems, no BAD citations are observed, reflecting the constraint that answers are generated only when at least one retrievable evidence passage is available. Pororoca exhibits a citation quality distribution closely aligned with the baselines, indicating that explicit abstention does not negatively affect attribution

Table 2: Distribution of citation quality for generated answers.

| System | GOOD (%) | WEAK (%) | BAD (%) |
|---|---|---|---|
| baseline_always | 79.8 | 20.2 | 0.0 |
| rag_no_abstention | 79.8 | 20.2 | 0.0 |
| pororoca | 78.9 | 21.1 | 0.0 |

Table 3: Abstention performance metrics.

| System | Accuracy | Precision | Recall |
|---|---|---|---|
| baseline_always | 0.759 | 0.375 | 0.161 |
| rag_no_abstention | 0.759 | 0.375 | 0.161 |
| pororoca | 0.770 | 0.364 | 0.071 |

quality when answers are produced. These results demonstrate that Pororoca preserves attribution fidelity while introducing stricter epistemic controls on when answers are allowed.

We evaluate abstention as a first-class decision problem. Table 3 reports abstention accuracy, precision, and recall, where the *positive class* is defined as no_evidence (i.e., a correct abstention is a true positive). Pororoca achieves the highest abstention accuracy (0.770), demonstrating improved alignment between evidence availability and generation decisions.

Abstention recall is low across all systems (0.071 for Pororoca), reflecting an intentionally conservative regime: the evidence threshold $\tau = 0.30$ was selected to minimise the risk of withholding answers to genuinely supported questions, which results in a system that abstains only when evidence is clearly insufficient. This trade-off is *by design*: in high-risk scientific QA, the cost of producing an unsupported answer outweighs the cost of abstaining on a borderline question. The risk–coverage curves in Figure 3 show that recall can be increased by raising $\tau$, at the controlled cost of lower coverage. Per-class decision counts (TP, TN, FP, FN) are reported in the confusion matrix in Appendix B (Table 4).

Importantly, abstention is treated as a correct outcome when evidence is insufficient, rather than as a failure, in line with the epistemic objectives of the system.

To explicitly analyze the trade-off between reliability and coverage, we plot selective risk–coverage curves by varying the evidence threshold $\tau$. Figure 3 shows that increasing $\tau$ monotonically reduces factual risk at the cost of lower coverage.

Pororoca exhibits a controlled and interpretable trade-off: at $\tau = 0.30$, the system reduces risk from 0.211 to 0.192 while maintaining coverage above 0.87. This behavior is unattainable for baselines lacking explicit decision thresholds, which collapse to single operating points. These results demonstrate that Pororoca enables principled tuning of epistemic risk, allowing deployment-specific calibration without retraining models. Taken together, the results show that Pororoca preserves factual quality when answers are generated, while substantially improving the system's ability to refrain from answering under insufficient evidence. The gains arise not from changes in model capacity or retrieval quality, but from the explicit enforcement of an evidence-gated decision policy.

These findings support the central thesis of this work: reliable scientific question answering is fundamentally a decision problem, and explicit abstention is a necessary component for aligning LLM-based systems with epistemological norms.

## 5  CONCLUSION

This work argued that reliable scientific question answering is not primarily a model-centric problem, but a system-level decision problem. In high-risk scientific domains, factual reliability requires not only grounded generation, but explicit mechanisms that determine when answering is epistemically justified and when abstention is the correct action. We introduced Pororoca, an evidence-gated scientific QA system that operationalizes this principle through a deterministic decision policy conditioning generation on verifiable evidence sufficiency. By enforcing explicit abstention and re-

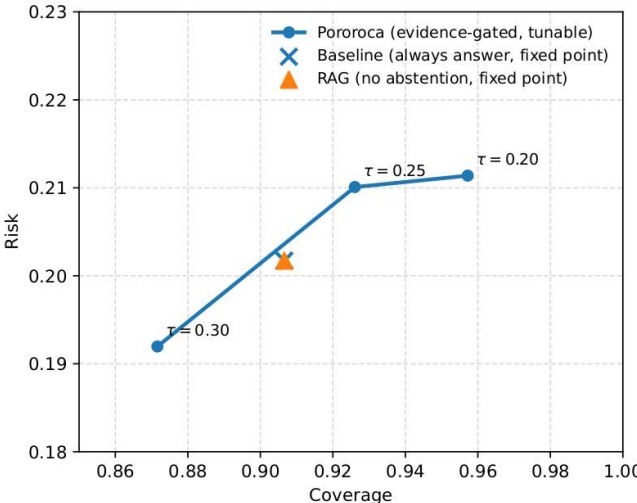

Figure 3: Selective risk–coverage trade-off for Pororoca and baselines. Coverage is plotted against factual risk as the evidence threshold $\tau$ varies. Pororoca exposes a tunable risk–coverage frontier, with explicit operating points annotated for each value of $\tau$. Baselines lacking explicit abstention mechanisms collapse to fixed operating points.

quiring document- and page-level provenance, Pororoca aligns language model outputs with core epistemological norms of scientific practice: traceability, verifiability, and auditability.

Our experimental results demonstrate that enforcing an evidence-gated decision policy does not degrade factual quality when answers are generated. Instead, it exposes a controllable risk–coverage trade-off that is inaccessible to conventional RAG pipelines lacking explicit abstention mechanisms. Crucially, the observed gains arise not from changes in model capacity, retrieval quality, or reranking, but solely from the explicit enforcement of epistemic decision rules. This confirms our central thesis that hallucination in scientific QA is fundamentally a decision problem rather than a generation problem. Beyond the specific system presented, this work contributes a methodological perspective on the design and evaluation of scientific QA systems. By treating abstention as a first-class, correct outcome under insufficient evidence, and by grounding evaluation in auditable execution logs, Pororoca provides a framework for studying reliability under realistic uncertainty and retrieval noise. The proposed evaluation protocol complements existing factuality and attribution benchmarks by explicitly incorporating selective behavior into system assessment.

Future work includes extending evidence-gated decision policies to additional scientific domains, evaluating performance under naturalistic query distributions, and integrating such policies with broader scientific workflows, including literature review, hypothesis exploration, and decision support. More broadly, we believe that explicit, auditable decision policies are a necessary foundation for deploying large language models as reliable tools in scientific contexts, where knowing when not to answer is as important as answering correctly.

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

## A  APPENDIX A: ANNOTATION AND CONSTRUCTION GUIDELINES FOR THE EVALUATION SET

This appendix provides a detailed description of the annotation protocol and the design principles used in constructing the evaluation set employed throughout this work. The purpose of this section is to make explicit the epistemic guarantees underlying the labels `has_evidence` and `no_evidence`, ensuring that the dataset is fully auditable and aligned with the decision-centric goal of scientific question-answering with evidence-gating.

### A.1  PURPOSE AND SCOPE

The evaluation set is not intended to assess the linguistic quality of the generated responses, nor to serve as a benchmark for overall performance in question answering. Instead, it was explicitly designed to evaluate a system's ability to decide whether answering a question is epistemically justified, given a fixed scientific corpus. Consequently, all questions in the dataset are well-formed, relevant to the domain, and scientifically meaningful. The only distinction between labels lies in the availability of explicit supporting evidence located in the corpus, at the document and page level.

### A.2  ROLE OF THE LANGUAGE MODEL IN DATASET CONSTRUCTION

An LLM (provided by OpenAI) is used exclusively as a linguistic generator of candidate questions. Its role is strictly limited to the production of questions in natural language, based on textual content extracted from the scientific corpus. Crucially, the language model is *not* used to:

- provide answers to questions;

- assess factual correctness;

- assign evidence labels;

- access external retrieval mechanisms, document collections, or evidence sources beyond the explicitly provided corpus excerpts.

While linguistic priors inherent to large language models cannot be fully eliminated, they do not affect evidence availability labels, which are assigned exclusively through human validation based on explicit corpus content.

This separation ensures that the evaluation set does not encode implicit knowledge from the model and that all labels are based solely on verifiable content from the corpus.

This procedure ensures that all generated questions are grounded in corpus content, while the subsequent human validation step verifies evidence availability independently of the generation process.

## A.3 DEFINITION OF EVIDENCE AVAILABILITY

Let $q$ be a scientific question and let $\mathcal{C}$ be the fixed scientific corpus processed by the Pororoca system.

A question $q$ is labeled as `has_evidence` if, and only if, there exists at least one textual excerpt $e \in \mathcal{C}$ such that:

1. $e$ is explicitly present in the corpus and is traceable to a specific document and page;

2. $e$ contains sufficient information to fully support a complete answer to $q$;

3. answering $q$ based on $e$ does not require external inference, extrapolation, or assumptions beyond what is explicitly stated.

On the other hand, a question $q$ is labeled as `no_evidence` if no textual excerpt of this nature exists in the corpus, even if the question is scientifically plausible, well-formulated, or commonly discussed in the literature.

It is important to note that `no_evidence` does *not* indicate that a question is invalid or unscientific. Rather, it indicates that answering the question would require knowledge not explicitly supported by the available corpus and, therefore, would violate the epistemic constraints imposed by evidence-gated generation.

## A.4 HUMAN VALIDATION PROTOCOL

All candidate questions generated by the language model undergo a human validation process conducted by three domain-trained annotators with backgrounds in meteorology and/or scientific information retrieval.

For each question, annotators:

1. inspect the relevant documents in the corpus;

2. check for the presence or absence of explicit supporting evidence;

3. check the location of the evidence at the document and page level;

4. assign a binary label (`has_evidence` or `no_evidence`) based solely on the availability of evidence.

**Inter-annotator agreement.** Each question was independently labeled by all three annotators. Inter-annotator agreement was measured using Cohen's $\kappa$ computed over all pairwise annotator combinations and then macro-averaged. The resulting $\kappa = 0.84$ indicates strong agreement, consistent with benchmarks reporting $\kappa > 0.80$ for well-specified binary annotation tasks (Rajpurkar et al., 2018).

**Adjudication and inclusion criteria.** Annotators are instructed to treat abstention (i.e., labeling a question as `no_evidence`) as a correct and desirable result whenever the evidence is insufficient. Questions for which it was not possible to reach unequivocal agreement (i.e., at least one annotator disagreed) were discarded. Only questions with full unanimous agreement (3/3) on the availability of evidence were included in the final evaluation set.

## A.5 DATASET BALANCING AND EXPERIMENTAL RATIONALE

The final dataset consists of 1020 questions, evenly balanced between the `has_evidence` and `no_evidence` instances.

This balance is intentional and serves a specific experimental purpose: to enable controlled evaluation of the system's epistemic decision-making capacity under equal prior probabilities. By decoupling the quality of the decision from the effects of class imbalance, we can directly assess the system's ability to discriminate between scenarios with and without sufficient evidence, independent of base rate effects.

Important methodological note: This balanced design prioritizes internal validity and interpretability of the abstention mechanism. In real-world scientific QA deployments, the distribution of answerable versus unanswerable questions would differ and depend on domain-specific factors, corpus coverage, and user query patterns. The balanced evaluation set should therefore be understood as a diagnostic benchmark for decision policy quality, rather than a simulation of operational deployment conditions. Future work should extend this evaluation to naturalistic query distributions to assess performance under realistic base rates.

The decision to prioritize controlled evaluation of the evidence-gating mechanism over ecological validity is justified by the following considerations: (i) establishing that the system *can* reliably distinguish evidence sufficiency is a necessary precondition before deployment, (ii) balanced evaluation enables direct interpretation of precision-recall trade-offs without confounding from class imbalance, and (iii) the threshold parameter $\tau$ can be adjusted post-deployment to accommodate different base rates and operational requirements. The resulting dataset is therefore optimized for the analysis of epistemic decision policies, rather than to reflect natural distributions of question frequency.

## A.6 REPRESENTATIVE EXAMPLES

Each instance in the dataset consists of a unique identifier, a question in natural language, and a binary evidence label. Examples are presented below.

```
{
  "id": "q736",
  "question": "Are atmospheric aerosol size distributions modeled using a universal
  "label": "has_evidence"
}

{
  "id": "q737",
  "question": "Does the aerosol model state that chemical composition has no influenc
  "label": "no_evidence"
}
```

In the first case, the corpus contains an explicit textual excerpt that describes the use of self-similar fractal fluctuations and inverse power law behavior in aerosol size spectra, providing sufficient support for a complete answer. In the second case, although the question is scientifically meaningful, no explicit statement in the corpus supports the proposition, and answering it would require unsupported extrapolation.

## A.7 IMPLICATIONS FOR EVIDENCE-GATED EVALUATION

By design, the evaluation set imposes a strict separation between linguistic plausibility and epistemic justification. As a result, the system's performance on this dataset directly reflects the quality of the

underlying decision policy governing generation versus abstention. This design ensures alignment between the evaluation protocol and the central objective of the Pororoca system: to avoid fluent but unsupported generation in scientific question-answering, conditioning responses on the existence of explicit and verifiable evidence.

## B    APPENDIX B: EXECUTION LOGS AND DECISION SEMANTICS

This appendix provides a detailed description of the execution logs generated by all systems evaluated in this work. The purpose of this section is to make explicit how system-level decisions to generate an answer or to abstain are recorded, interpreted, and used as the sole source of truth for experimental evaluation. By documenting the log structure and its semantics, this appendix ensures that the evaluation protocol is fully auditable, reproducible, and aligned with the decision-centric objective of evidence-gated scientific question answering.

### B.1    PURPOSE AND SCOPE

The execution logs are not intended to expose low-level implementation details or internal model states. Instead, they serve as an epistemic trace of the system's behavior at inference time, recording the availability of evidence, the applied decision policy, and the resulting action. All experimental results reported in the main paper are computed exclusively from these logs.

Each log entry corresponds to a single question evaluated under a specific system configuration. Across systems, the document corpus, retrieval mechanism, reranking procedure, and language model remain fixed. The only variation recorded in the logs concerns the decision policy governing whether answer generation is authorized or explicitly suppressed through abstention.

### B.2    EVALUATED SYSTEM MODES

Three system modes are evaluated and explicitly recorded in the execution logs:

- `baseline_always`, corresponding to a traditional retrieval-augmented generation pipeline that always produces an answer, regardless of evidence sufficiency;
- `rag_no_abstention`, corresponding to a RAG configuration in which retrieval and reranking are performed, but no explicit abstention mechanism is available;
- `pororoca`, the proposed evidence-gated system, which explicitly decides between answer generation and abstention based on the sufficiency of retrieved scientific evidence.

These modes should be interpreted as distinct *decision policies*, not as distinct language models.

### B.3    REPRESENTATIVE EXECUTION LOG

A representative execution log for a single evaluation instance is shown below. For clarity, the example is minimally reduced to include only the fields relevant to epistemic decision-making and evaluation.

```
{
  "run_id": "run_20260205_001324",
  "qid": "q817",
  "system": "pororoca",
  "label": "has_evidence",
  "decision": "ANSWER",
  "evidence_count": 3,
  "evidence_score": 0.81,
  "threshold": 0.75,
  "citations": [
    {"doc_id": "D12", "page": 4},
    {"doc_id": "D12", "page": 5}
  ]
```

```
}
```

Equivalent log entries are produced for the same question identifier (`qid`) under the `baseline_always` and `rag_no_abstention` system modes, differing only in the value of the `system` and `decision` fields.

### B.4 Definition of Log Fields

Each field in the execution log has a precise and interpretable meaning:

- `run_id` uniquely identifies the experimental run and groups together all system executions performed under the same experimental conditions.
- `qid` uniquely identifies the evaluation question and enables paired comparison across different system modes.
- `system` denotes the decision policy under evaluation and takes one of the values `baseline_always`, `rag_no_abstention`, or `pororoca`.
- `label` represents the ground-truth evidence availability assigned during dataset construction, taking values `has_evidence` or `no_evidence`.
- `decision` records the epistemic action taken by the system. `ANSWER` indicates that answer generation was authorized, whereas `NO_ANSWER` indicates explicit abstention due to insufficient evidence.
- `evidence_count` indicates the number of distinct document–page excerpts used as supporting evidence.
- `evidence_score` corresponds to the aggregated similarity score $s(q)$ computed from the retrieved evidence.
- `threshold` records the sufficiency threshold $\tau$ applied by the decision policy.
- `citations` list the document identifiers and page numbers associated with the evidence used to support the generated answer, enabling direct human verification.

### B.5 Relation to Abstention and Evaluation Metrics

The execution logs provide a complete and explicit record of when and why the system generates an answer or abstains. Abstention is treated as a correct and desirable outcome in scenarios labeled as `no_evidence`, reflecting adherence to epistemic constraints rather than system failure.

All metrics reported in the main paper are derived directly from the execution logs. Evidence-based factuality and citation quality are computed from the relationship between generated answers and logged citations. Abstention accuracy, precision, and recall are computed by comparing the `decision` field against the ground-truth `label`. Selective risk and coverage curves are obtained by varying the logged `threshold` parameter and recomputing metrics over the corresponding subsets of answered questions.

### B.6 Implications for Evidence-Gated Evaluation

By design, the execution logs enforce a strict separation between linguistic plausibility and epistemic justification. They provide an auditable trace of system behavior at the level of decision policy, ensuring that all reported results can be independently verified. This design aligns the experimental protocol with the central objective of the Pororoca system: to condition scientific question answering on the existence of explicit and verifiable evidence, and to avoid fluent but unsupported generation through explicit abstention.

### B.7 Confusion Matrix

Table 4 reports the per-class decision counts for all evaluated systems, where the positive class is `no_evidence` (a true positive is a correctly withheld answer; a false negative is an incorrectly generated answer in the absence of evidence).

Table 4: Confusion matrices for all systems (positive class = no_evidence). TP = correctly abstained; TN = correctly answered; FP = incorrectly abstained; FN = incorrectly answered when evidence was absent.

| System | TP | TN | FP | FN |
|---|---|---|---|---|
| baseline_always | 82 | 695 | 0 | 243 |
| rag_no_abstention | 82 | 695 | 0 | 243 |
| pororoca | 36 | 749 | 46 | 189 |

