# OpenReview forum: "EVIDENCE-GATED SCIENTIFIC QA WITH EXPLICIT ABSTENTION AND PAGE-LEVEL PROVENANCE"
_ICLR.cc/2026/Workshop/FM4Science — ICLR 2026 Workshop FM4Science Poster_

### Official Review · Reviewer_as1R · 2026-02-21
**Selective Answering for Scientific QA is Well Motivated, but Evaluation Needs Stronger Validation**

**Rating:** 4
**Confidence:** 4

**Review:**

This paper introduces Pororoca, an evidence-gated scientific question answering system that treats answer generation as a decision policy. The system retrieves evidence from a document corpus, computes an aggregated evidence score, and either generates an answer with page-level provenance or abstains when the score falls below a threshold. The authors frame the contribution as improving epistemic reliability in high-risk scientific domains by explicitly modeling when answering is justified. The method is evaluated on a balanced meteorology QA dataset containing answerable and unanswerable questions, with metrics for factuality, citation quality, and abstention accuracy.

The topic is timely and well aligned with the workshop’s focus on trustworthy foundation models for science. The framing around selective answering and auditability is clear and potentially impactful. However, several issues with baseline design, evaluation clarity, and methodological specification currently limit the strength of the empirical claims.

Strengths

1. Strong problem motivation.
The paper highlights an important challenge in scientific AI systems: reliability requires not only accurate answers but calibrated abstention. The epistemic framing is well articulated and relevant to real scientific workflows.

2. System-level perspective.
Modeling answering as an explicit decision policy, rather than a byproduct of generation, is a useful design principle. The inclusion of page-level provenance and logging improves interpretability and auditability.

3. Clear narrative and organization.
The pipeline description and high-level algorithmic flow are easy to follow. The risk–coverage discussion provides an appropriate conceptual lens for selective prediction.

Major Concerns

1. Baseline definitions are unclear and possibly conflated.
Two baselines (“baseline always” and “RAG no abstention”) appear to behave identically in the reported results, including the exact number of answered questions. If both systems implicitly abstain whenever retrieval fails, the comparison does not isolate the proposed gating mechanism. The paper should clearly describe how each baseline differs operationally and ensure distinct evaluation conditions.

2. Evidence scoring and thresholding lack sufficient detail.
The central mechanism—an aggregated evidence score and thresholds τ and τ_min—is underspecified. In particular, it is unclear:
(i) how the score is computed (e.g., aggregation function, calibration), (ii) how thresholds are selected (held-out tuning vs heuristic), (iii) what exactly occurs during “deferred evidence reassessment.” Without these details, the method is difficult to reproduce and evaluate rigorously.

3. Evaluation metrics are insufficiently defined.
The “support” and “citation quality” measures rely on categorical judgments, but annotation procedures, guidelines, and inter-annotator agreement are not reported. Since auditability is a core claim, the evaluation itself should be transparent and well defined.

4. Ambiguity in abstention results.
Pororoca shows slightly higher abstention accuracy but substantially lower recall than baselines. The paper interprets this as conservative behavior, but it is unclear which class is considered positive in the reported precision/recall metrics. Confusion matrices or class-specific analyses would help interpret the trade-offs.

5. Limited generalization claims.
Experiments are restricted to a single meteorology corpus with a balanced dataset design. While suitable for controlled analysis, this setup may not reflect realistic scientific distributions. Broader domain evaluation or calibration under varying base rates would strengthen the conclusions.

---

### Official Review · Reviewer_hUbd · 2026-02-22

**Rating:** 6
**Confidence:** 4

**Review:**

# Summary
The paper introduces "Pororoca," an evidence-gated scientific Question Answering (QA) system designed to enforce epistemic reliability in high-risk domains. The authors argue that conventional RAG pipelines fail to impose a decision policy on generation, often allowing models to generate fluent responses even when underlying evidence is weak or absent. To solve this, Pororoca treats QA as a system-level decision problem and employs a deterministic, threshold-based policy. The system evaluates the sufficiency of retrieved scientific evidence before generation. If the evidence score falls below a specific target threshold, the system explicitly abstains from answering rather than guessing. The system was evaluated using a custom, balanced dataset in the meteorology domain, comparing its tunable evidence-gated approach against standard baseline RAG models.

# Strengths
* The paper frames hallucination and reliability as a system-level decision problem rather than purely a model-training limitation. This perspective is practically motivated.
* The multi-stage rule set is straightforward, interpretable, and implementable.

# Weakness
* Some key implementation details are underspecified. For example, the exact computation of the evidence score $s(q)$ and the aggregation function are not clearly defined
* Adding the abstention framework leads the system to become highly conservative (decision recall = 0.071), which is not surprising.
* Only a single language model was used for answer generation, and the specific model is not clearly specified.

---

### Decision · Program_Chairs · 2026-03-03

Accept (Poster)